# The Role of Nutrition in the Pathogenesis and Treatment of Autoimmune Bullous Diseases—A Narrative Review

**DOI:** 10.3390/nu16223961

**Published:** 2024-11-20

**Authors:** Aleksandra Anna Kajdas, Agnieszka Żebrowska, Anna Zalewska-Janowska, Aneta Czerwonogrodzka-Senczyna

**Affiliations:** 1Department of Clinical Dietetics, Medical University of Warsaw, Erazma Ciolka 27 Street, 01-445 Warsaw, Poland; aneta.czerwonogrodzka@wum.edu.pl; 2Department of Dermatology and Venereology, Medical University of Lodz, Hallera sq. 1, b. no. 6, 90-647 Lodz, Poland; agnieszka.zebrowska@umed.lodz.pl; 3Psychodermatology Department, Medical University of Lodz, 251 Pomorska Street, 92-213 Lodz, Poland; anna.zalewska-janowska@umed.lodz.pl

**Keywords:** autoimmune bullous diseases, nutrition, diet, pemphigus, pemphigoid, dermatitis herpetiformis, epidermolysis bullosa acquisita

## Abstract

Autoimmune bullous diseases (AIBDs) are a group of conditions marked by the formation of blisters and erosions on the skin and mucous membranes. It occurs in all age groups, slightly more often affecting women. Several factors may be linked to the development of AIBDs, with nutrition being one of them. The literature mentions various food products and food ingredients acting as disease modifiers. Given the complex relationship between bullous diseases and nutrition, the current literature on AIBDs has been reviewed, with an emphasis on the influence of dietary modifications, various diets, and the nutritional consequences of these conditions. This review summarizes the role of nutrition in the pathogenesis and treatment of the following AIBDs: (i) pemphigus, (ii) bullous pemphigoid and mucous membrane pemphigoid, (iii) dermatitis herpetiformis, and (iv) epidermolysis bullosa acquisita. Several nutrients and dietary factors have been studied for their potential roles in triggering or exacerbating AIBDs. The key nutrients and their potential impacts include thiols and bulb vegetables (*Allium*), phenols, tannic acid, tannins, phycocyanin, isothiocyanates, all trans-retinoic acids, cinnamic acid, and walnut antigens. Many patients with ABIDs may require supplementation, particularly of vitamin D and B_3_, calcium, potassium, zinc, selenium, and cobalt. In addition, various diets play an important role. A soft diet is recommended for individuals with issues in the oral cavity and/or esophagus, particularly for those who experience difficulties with biting or swallowing. This approach is commonly used in managing pemphigus. A high-protein, high-calcium diet, DASH (Dietary Approaches to Stop Hypertension), and the Mediterranean diet are utilized during long-term glucocorticoid therapy. However, in dermatitis herpetiformis it is advisable to follow a gluten-free diet and eliminate iodine from the diet. When it comes to herbal supplements, Algae (*Spirulina platensis*), *Echinacea*, and St. John’s wort (*Hyperitum perforatum*) enhance the ABIDs, while *Cassia fistula* may be recommended in the treatment of erosions in pemphigus vulgaris. Fast foods enhance the development of ABIDs. However, the pathomechanism is not yet fully understood. Future researchers should more precisely define the relationships between nutrients and nutrition and blistering diseases by also looking at, i.e., genetic predispositions, microbiome differences, or exposure to stress.

## 1. Introduction

Autoimmune bullous diseases (AIBDs) are a group of conditions marked by the formation of blisters and erosions on the skin and mucous membranes [1]. These diseases arise when the body’s immune system erroneously targets proteins crucial for maintaining skin integrity, leading to the separation of skin layers and blister formation [2]. It occurs in all age groups, and slightly more often affects women. To date, AIBDs have been classified into different types and subtypes based on clinical, histopathological, and immunological characteristics [3]. Most of them are severe and resistant to treatment. As such, they often require a range of therapies, including steroids, immunosuppressants, or biologics [4]. There are several factors that may be attributed to the development of AIBDs, such as genetic predisposition, medications, infections, hormonal changes, chronic stress, or other autoimmune diseases [5]. Food is not commonly considered as a direct cause of AIBDs, however, in some cases, certain foods can act as triggers or aggravating factors, particularly in individuals who have other autoimmune conditions. An appropriate diet is usually a method that supports treatment [6].

Pemphigus occurs in all age groups, with its peak incidence around the age of 50–60, and it affects women slightly more often. The most common variety is pemphigus vulgaris [7]. Pemphigus is a potentially life-threatening disease. Exogenous factors that may cause pemphigus include drugs, UV radiation, viral infections, and vegetables from the *Allium* group containing sulfhydryl groups—garlic, leek, and onion [8]. Pemphigoid usually occurs in older people, over 60 years of age, with up to a 30-fold increase in patients over 80 years of age compared to 60-year-olds. Most patients also suffer from internal diseases, such as diabetes, hypertension, and coronary artery disease, and some of them have a history of stroke or heart attack. Due to numerous concomitant diseases, the treatment of pemphigoid should be supervised by a dermatologist experienced in the treatment of bullous diseases, working in a reference center, or cooperating with such a center. The group of specialists involved in the patient’s therapy should include, depending on their needs, a district dermatologist, a family doctor or a geriatrician, a neurologist, a nurse experienced in caring for the elderly, a dietician, and a psychologist. An appropriate diet is usually a method that supports treatment. Some patients report certain products as a trigger or making the disease worse (thiols in onion vegetables, phenols contained in mangoes and pistachios, tannins, cinnamic acid, walnut antigens, and some herbs) [9]. Dermatitis herpetiformis is the only bullous disease in which a proper diet is important in the treatment of the patient. In this disease, the autoimmune process towards transglutaminases (TGs) in the skin is usually accompanied by clinically asymptomatic or oligosymptomatic, gluten-sensitive enteropathy (GSE). Some patients feel a significant improvement when strictly following a gluten-free diet, and some do not even need pharmacological treatment. The combination of pharmacological treatment with dapsone and the use of a gluten-free diet in individualized combinations is the therapeutic management of choice in dermatitis herpetiformis [10]. The incidence of epidermolysis bullosa acquisita (EBA) occurs in both children and adults of all ages [11]. The disease affects the skin and mucous membranes [12]. The relationship between nutrition and AIBDs is multifaceted [13]. The literature mentions various food products and food ingredients acting as disease modifiers [14]. Among various nutritional groups, the following may act as triggers or stop the development of the disease: nutrients, vitamins and minerals, diets, herbal supplements, and fast foods. They have been described in detail in the later parts of the manuscript. This review summarizes the complex relationships between AIBDs and nutrition with an emphasis on the influence of dietary modifications, various diets, and the nutritional consequences of these conditions. It focuses on the following ABIDs: (i) pemphigus, (ii) bullous pemphigoid and mucous membrane pemphigoid, (iii) dermatitis herpetiformis, and (iv) epidermolysis bullosa acquisita.

## 2. Characteristics of Autoimmune Bullous Diseases

In the following paragraphs, the four groups of AIBDs are discussed, covering their epidemiology, etiopathogenesis, clinical presentation, and diagnosis, as well as their pharmacological treatment. The summary is described in Table 1.

### 2.1. Pemphigus

In Europe, the incidence of pemphigus is 0.5–8/1,000,000/year; it occurs in all age groups with the peak incidence around the age of 50–60, and it affects women slightly more often. The most common variety is pemphigus vulgaris (70–80% of cases in Europe, the USA, and Japan) [7]. Pemphigus is an autoimmune disease which is potentially life-threatening, in which the immune system is dysregulated and autoantibodies are formed against desmogleins, i.e., surface proteins of keratinocytes, namely desmoglein 3 and/or 1. The binding of these autoantibodies to desmogleins causes acantholysis (cleavage of intercellular junctions of the epidermis). An association with HLA antigens has been described. Exogenous factors that may cause pemphigus include drugs—mostly containing a thiol group (-SH), but also non-thiol drugs, e.g., penicillamine, angiotensin-converting enzyme inhibitors, calcium channel blockers, penicillins, cephalosporins, and non-steroidal anti-inflammatory drugs. Other triggering factors were described, for example burns, UV radiation, viral infections, and vegetables from the *Allium* group containing sulfhydryl groups—garlic, leek, and onion [8].

Pemphigus occurs in two main classic varieties—pemphigus vulgaris and foliaceus. The third variety is paraneoplastic pemphigus, which coexists with cancer, mainly non-Hodgkin’s lymphoma and chronic lymphocytic leukemia. Pemphigus vulgaris is a potentially life-threatening disease due to the development of epidermal barrier disorders resulting in severe water and electrolyte disturbances and extensive secondary microbial infections. The introduction of glucocorticosteroids (GCSs) into treatment in the second half of the 20th century significantly reduced the mortality rate due to this disease from 75% to 30% [8].

Pemphigus vulgaris initially manifests itself in 70–90% of cases with mucosal lesions in the form of erosions resulting from easily bursting superficial blisters. Erosions are most often located on the mucous membranes of the oral cavity, less often also on the throat and the initial sections of the respiratory system, conjunctiva, or the epithelium of the urogenital system. Patients often complain of swallowing problems and hoarseness. Usually, a few weeks after the appearance of mucosal lesions, blistering lesions appear on the skin and/or scalp in the form of flaccid blisters that rupture easily into erosions. These, in turn, become covered with scabs and heal, leaving discoloration and without scarring [15].

The clinical picture of pemphigus foliaceus is dominated by very superficial erosions covered with crusts and exfoliation. Due to their very superficial location in the epidermis, blisters are usually not present on the skin, only crusting and weeping erosions. The lesions are located mainly on the upper chest and back, especially in the seborrheic areas, and on the face and scalp. The lesions do not affect the mucous membranes [8].

Paraneoplastic pemphigus, as it coexists with cancer and has a significant predilection for affecting all mucous membranes, requires a special description. Erosions in the upper gastrointestinal tract often lead to the development of ulcerative stomatitis and swallowing disorders, which require a specialized dietary approach, which will be discussed in detail below. Less common types of pemphigus, for example, IgA pemphigus, that do not require the implementation of different procedures in the patient’s diet therapy, will not be described in this chapter to maintain the clinical clarity of autoimmune bullous diseases [15,16,17].

The diagnosis of pemphigus is based on the clinical picture of the lesions, the histopathological picture of skin and/or mucous membrane sections, and the results of immunological tests, but the basic diagnostic test is the examination of a skin section using the direct immunofluorescence method, in which deposits of antibodies are observed, usually of the IgG class with the complement component C3 or without it, which are located in the intercellular spaces of the epidermis and epithelium. Serum testing using the indirect immunofluorescence method is used to determine the titer of pemphigus antibodies in the patient’s serum. The concentration of anti-desmoglein 1 antibodies shows a closer correlation with the patient’s clinical condition than anti-desmoglein 3 antibodies. Rituximab, an antibody directed against the CD-20 molecule, is currently the first-line drug in the treatment of paraneoplastic pemphigus. It is used as monotherapy or in combination with systemic glucocorticoids, which are classic drugs used in the treatment of pemphigus. GCSs combined with immunosuppressive drugs, such as azathioprine or cyclophosphamide, are the basic therapy for other forms of pemphigus. Local treatment in the form of disinfectant and anti-inflammatory preparations is always used in the case of active lesions [18,19].

### 2.2. Bullous Pemphigoid and Mucous Membrane Pemphigoid

Bullous pemphigoid (BP) is the most common autoimmune bullous disease, typically occurring in elderly people over 60 years of age, with no gender predilection. The incidence is estimated at 0.2–4 cases/100,000/year. Patients with pemphigoid often suffer from neurological and psychiatric diseases. Sometimes, pemphigoid is paraneoplastic syndrome. Literature data indicate the possible occurrence of cross-immune reactions between skin and central nervous system antigens. The antigens in this autoimmune disease are hemidesmosomal proteins within the dermal–epidermal junctions with a molecular weight of 180 kD (BP180 antigen) and 230 kD (230BP antigen). When autoantibodies combine with antigens, the inflammatory cascade involving neutrophils, eosinophils, and complement components is activated, which leads to damage to the dermal–epidermal junctions. Exogenous factors that may cause pemphigoid include the following: UV radiation; mechanical and thermal injuries; stress; viral infections; and drugs—gliptins, PD1 inhibitors, furosemide, amoxicillin, ciprofloxacin, potassium iodide, gold salts, and spironolactone. Mucous membrane pemphigoid (MMP), formerly called scarring pemphigoid, is a rare type of pemphigoid affecting the mucous membranes—in 85% of cases the mucous membranes of the oral cavity are affected—and only in 20–30% of cases, the skin [9].

Typical lesions are subepidermal tense blisters, on erythema and urticarial skin and on normal-looking skin. They are most often located on the skin of the trunk and the flexural surfaces of the limbs. Mucous membranes are affected in 10–20% of cases and most often affect the oral cavity, where the disease is dominated by painful erosions. The lesions are usually accompanied by skin itching. It is emphasized that in each case of severe pruritus in elderly patients, the diagnosis of pemphigoid should be considered. In mucous membrane pemphigoid, lesions may subside with scarring. Atypical clinical forms of pemphigoid are not the subject of this study [20].

Pemphigoid is diagnosed based on the clinical picture, histopathological examination, and immunopathological test results. Histopathological examination reveals a subepidermal blister and an inflammatory infiltrate with a predominance of eosinophils located around blood vessels. The basis of diagnosis is the detection of linear deposits of IgG and/or complement component C3 along the basement membrane in direct immunopathological examination. Often, indirect immunopathological examination, i.e., of the patient’s serum, detects circulating antibodies against pemphigoid-typical antigens BP180, BP320, or other antigens. The concentration of antibodies does not correlate with the activity or severity of the disease. The first-line treatment is strong local glucocorticoids (clobetasol propionate), which are used for a few months after the lesions disappear. Systemic glucocorticoids are also effective in the treatment of this disease; however, due to the older age of patients, many comorbidities, and increased exposure to the development of side effects, experts advise against this method of treatment as the first line. Other drugs used in the treatment of pemphigoid include methotrexate, tetracyclines, azathioprine, mycophenolate mofetil, cyclophosphamide, and dapsone [9,21].

### 2.3. Dermatitis Herpetiformis (Duhring’s Disease)

Dermatitis herpetiformis (Duhring’s Disease) is a chronic, intensely itchy skin disorder characterized by clusters of small blisters and red, raised patches, typically appearing on the elbows, knees, buttocks, and scalp [22]. The incidence of Duhring’s disease is 0.4–3.5/100,000/year. It decreases by 4% annually, which may be due to the preventive effect of a gluten-free diet in patients with gluten-dependent enteropathy. The disease affects men slightly more often than women, and the onset of changes usually occurs in the 4th or 5th decade of life; it may run in families, more often in siblings than in parents and children. In approximately 90% of patients, features of gluten-dependent enteropathy are detected in histopathological sections from the mucosa of the small intestine. Both genetic factors—most patients have HLA DQ2 and HLA DQ8—and environmental factors play a role in the etiopathogenesis of the disease. The main autoantigen is tissue transglutaminase. In people with a genetic predisposition, the immune response is stimulated and the synthesis of specific IgA directed against tissue transglutaminase is triggered, and the autoimmune process is extended to include IgA recognition of epidermal transglutaminase antigens. In histopathology and immunopathology, this phenomenon is seen as complexes consisting of IgA and tissue transglutaminase in the apex of dermal papillae. This mechanism leads to the development of changes typical of dermatitis herpetiformis. It is worth emphasizing that a diet rich in gluten increases the immune reaction against tissue transglutaminase. It has been observed that the use of a gluten-free diet in Duhring’s disease reduces IgA synthesis by 50% after 2 months of its use [10,22].

Polymorphic skin lesions in the form of vesicles, blisters, exudative papules, urticarial eruptions, and erosions covered with crusts, often with a herpetic pattern, are symmetrical and most often affect the skin of the elbows and knees (90% of cases) and the skin of the buttocks and the lumbosacral area. The skin in other areas may also be affected. Mucosal involvement is very rare. The skin lesions are accompanied by severe itching and burning of the skin. The autoimmune process towards skin transglutaminases is usually accompanied by clinically asymptomatic or mildly symptomatic gluten-dependent enteropathy. Fewer than 10% of patients have malabsorption syndrome, flatulence, or diarrhea. Intensification of skin lesions is also observed after exposure to iodine, both taken with the diet and applied externally [23,24].

The diagnosis of dermatitis herpetiformis is based on the characteristic clinical picture of the lesions, confirmed by immunological tests and possibly by histopathological examination. The diagnosis is based on the detection of granular IgA deposits in skin papillae during direct immunofluorescence examination of a section of unaffected skin near the lesions. The gold standard of treatment is a gluten-free diet and in some cases, a combination of diet and dapsone. Symptoms such as itching and burning of the skin disappear spectacularly after 24–48 h of using the drug. Following this, the skin lesions slowly disappear. Dapsone does not affect intestinal changes. At the same time as the introduction of dapsone, a gluten-free diet should be implemented, which affects both the skin and intestinal lesions, and its effects are visible after at least 3–6 months. A gluten-free diet also reduces the risk of developing gastrointestinal lymphomas in patients with dermatitis herpetiformis. Exacerbation of the disease is also observed after exposure to iodine. Patients should avoid iodine in food (sea fish, seafood, sea algae, iodized salt, mineral waters containing iodine) and external exposure, i.e., disinfectants with iodine or iodine in seaside air [25,26].

### 2.4. Epidermolysis Bullosa Acquisita

This is an autoimmune subepidermal bullous disease in which the autoimmunity process is directed against epitopes of collagen VII, which is a protein that forms anchoring fibers connecting the basement membrane with the dermis. The incidence of epidermolysis bullosa acquisita (EBA) is estimated at 0.25/1,000,000 and occurs in both children and adults of all ages. Inflammatory bowel disease is diagnosed in 25% of patients. Adult patients may have comorbid cancer [11].

The disease affects the skin and mucous membranes. Usually, blisters are located on the extensor parts of the limbs, which are susceptible to mechanical injuries. The lesions heal, leaving milia and atrophic scars. Mucous membranes are dominated by erosions after rupture blisters, especially in the mouth and the mucous membranes of the eyes and genitals [13].

The diagnosis is made based on the clinical picture and additional tests. Direct immunofluorescence examination reveals IgG deposits arranged linearly along the basement membrane. Indirect immunofluorescence tests detect the presence of circulating IgG antibodies that bind to the basement membrane. To differentiate it from pemphigoid, a skin split test is performed. In pemphigoid, immune deposits are located in the roof of the artificially created blister, while in EBA, they are located in the dermal part of the blister. The histopathological results are inconclusive and may resemble pemphigoid. Pharmacotherapy is based on systemic glucocorticoids, sometimes in combination with sulfones. Other options include colchicine, immunosuppressive drugs, or immunoglobulin infusions [27].

## 3. The Role of Nutrition in Autoimmune Bullous Diseases

The key findings of the role of nutrition, including nutrients, vitamins and minerals, diets, herbals supplements, and fast foods, in the pathogenesis and treatment of ABIDs have been presented in Table 2.

### 3.1. Nutrients

#### 3.1.1. Thiols and Bulb Vegetables (*Allium*)

Thiols are organic compounds that contain a sulfhydryl functional group (-SH) [28]. *Allium* vegetables contain both thiol and disulfide groups. These compounds play a role in the formation of blisters, especially in patients with a genetic predisposition. Thiols are found in food (garlic, onion, leek, and chives), medicines (captopril and penicillamine) and cosmetic products. Sulfhydryl radicals inhibit the enzymes that aggregate keratinocytes and activate the enzymes that disaggregate them. Additionally, thiols form thiol–cysteine bonds instead of cysteine–cysteine bonds, which further disturbs cell adhesion. According to the latest scientific research, 15% of patients reported garlic as a trigger for pemphigus. Many patients declared that their symptoms were exacerbated and/or triggered after consuming leeks and garlic, and they declared that their symptoms disappeared after eliminating garlic, onions, and leeks from their diet [29,30,31].

#### 3.1.2. Phenols

Phenols are organic compounds whose hydroxyl group is attached to a carbon that is part of the aromatic ring. This group of compounds includes free phenols (which are often components of essential oils), and phenolic glycosides and phenyl acids (which are components of plants). These compounds are found in the following foods: pistachio, cinnamon, mango, red pepper, black pepper, thymol (essential oil of thyme), eugenol (essential oil of clove flower buds), aspartame, cinnamon bark oil, pea seeds, rosmarinic acid, cynarin (artichoke), allspice, fennel, curcumin (Curcuma longa root), arbutin (viburnum leaves, pear leaves, and marjoram herb), mace, and milk. Studies have shown that keratinocytes exposed to phenol release IL-1α and TNFα, which increase complement and protease synthesis, thereby contributing to inflammation and acantholysis in tissue samples from pemphigus vulgaris patients and mice. These compounds are involved in the induction of pemphigus in genetically predisposed patients. However, it should be strongly emphasized that the elimination of phenolic compounds from the diet absolutely does not allow any reduction in the dose of systemic glucocorticoids in these patients. Phenols are also found in cleaning products. A case of pemphigus vulgaris has been described in a woman after contact with cleaning products containing phenols [2,32,33,34,35,36].

#### 3.1.3. Tannic Acid

Tannic acids are a group of organic chemical compounds composed of D-glucose and gallic acid. They are found in coffee, tea, eggplant, cassava, cherries, blackberries (leaves), cranberries, ginger, avocados, oak bark, blueberries, wild strawberries, sage, willow bark, walnuts, cashew nuts, rosemary, ground pepper (betel), cassava, and mango. After incubation of keratinocyte cultures with various concentrations of gallic acid, a key role of the local immune response was found. Tannins play a special role in the case of endemic pemphigus. Endemic pemphigus foliaceus (*fogo selvagem*) has been reported in the region in the state of Mato Grosso do Sul in Brazil, where the prevalence of pemphigus foliaceus during the mid-1990s was around 3% [33]. An endemic form of pemphigus vulgaris has also been reported in a small number of patients residing in an endemic region of pemphigus foliaceus in Brazil [34]. The presence of tannins in the food and living environment of people in the Amazon basin, India, and Brazil is suspected, e.g., due to the exceptionally high consumption of Indian tea, water from the Amazon basin, and guarana—popular in South America. These compounds are involved in the induction of pemphigus, especially in genetically predisposed patients. Importantly, eliminating tannins from the diet absolutely does not allow for reducing the dose of systemic glucocorticoids in these patients [37,38,39,40,41,42].

#### 3.1.4. Tannins

Tannins are polyphenolic compounds naturally found in tree bark, plants, black pepper, cherries, blueberries, mangoes, cashews, tea, vanillin, and cocoa. They are capable of forming complexes with metal ions. Tannins are also believed to have antioxidant properties. Research shows that patients who consume large amounts of tannins in their diet also have higher concentrations of them in their skin. These communities are suspected to be disproportionately affected by the disease due to their proximity to rivers with high tannins in their water systems and diet. Additionally, the diagnosis rate of pemphigus vulgaris is high in India, which may be related to the consumption of large amounts of tea and betel nut [43].

#### 3.1.5. Phycocyanin and Isothiocyanates

Phycocyanin is a blue pigment found in cryptophytes, cyanobacteria, and red algae. It has fluorescent and antioxidant properties. This compound is responsible for the induction of pemphigus, especially in genetically predisposed patients. The case of a 57-year-old man has been reported with pemphigus vulgaris whose symptoms worsened after consuming a diet containing phycocyanin. However, again, the elimination of phycocyanin from the diet does not allow for any modification of the dose of systemic glucocorticoids in these patients [44,45].

Isothiocyanates (so-called mustard oils) are compounds belonging to glucosinolates that contain glycosidic sulfur links. They are generally unstable compounds—liquid and oily. They are found in white and black mustards, as well as in horseradish and nasturtium. They are responsible for the induction of pemphigus, especially in genetically predisposed patients. Despite reports regarding the association of isothiocyanate with causing blistering lesions on the oral mucosa, there are no confirmed studies on this subject so far. As in the case of tannins, phenols, and phycocyanin, eliminating isothiocyanates from the diet does not allow for reducing the dose of systemic glucocorticoids in patients [45].

#### 3.1.6. All Trans-Retinoic Acids and Cinnamic Acid

The vitamin A metabolite, retinoic acid, has been shown to modulate the immune system by influencing the proliferation, differentiation, and apoptosis of immune cells. These metabolites may play a role in modulating pemphigus by causing Th17 cell depletion while stabilizing regulatory T cells. This relationship is further complicated by the fact that all trans-retinoic acids shift the balance of the Th1 to Th2 ratio towards Th2, which is known to be elevated in patients with pemphigus [20].

Cinnamic acid is an organic chemical compound containing a double bond and a benzoic ring. It is found in candied fruit, tomatoes, oranges, and grapefruits. In one study in which pemphigus patients were asked to list possible triggers, the most frequently mentioned food-related trigger was tomato (23.1%), which is high in cinnamic acid [46].

#### 3.1.7. Walnut Antigens

Walnut antigens introduced into the body cause an immune reaction consisting of the proliferation of lymphocytes and the formation of specific antibodies. Research suggests that exposure to walnut antigen through gastrointestinal epithelial cells may activate B cells in individuals genetically predisposed to pemphigus vulgaris through a hit-and-run mechanism. By this mechanism, cross-reactivity between the infectious antigen and the self-antigen may lead to a long-lasting immune response, even after the pathogen has been cleared, because the continued presence of the self-antigen will continually drive the generation of autoantibodies and the development (as well as perpetuation and/or exacerbation) of the disease [47,48].

### 3.2. Vitamins and Minerals

#### 3.2.1. Vitamins D and B_3_

Vitamin D includes vitamin D_2_ (ergocalciferol) and vitamin D_3_ (cholecalciferol). It is of fundamental importance in the regulation of calcium and phosphate metabolism and the metabolism of bone tissue. The most important sources of vitamin D for humans include its endogenous synthesis (approx. 80%). This vitamin is found mainly in products of animal origin (oils, fatty fish, and eggs). There are several hypothetical roles of vitamin D in the pathogenesis of pemphigus vulgaris. In vitro studies have shown that vitamin D increases beta cell apoptosis while reducing their proliferation and increasing the number and function of regulatory T cells. Vitamin D has also been shown to protect keratinocytes from several apoptosis pathways. It was found that patients with pemphigus vulgaris had lower vitamin D levels, regardless of body mass index, age, and sun exposure. Its supplementation is important in the case of pemphigus treatment with the use of chronic glucocorticosteroid therapy and during long-term treatment of pemphigus with the use of immunosuppressive drugs [49,50,51,52,53,54].

Vitamin B3 is the general name for two compounds: niacin and nicotinamide. It acts as a precursor for NADP and NAD coenzymes involved in oxidation and reduction processes. It is commonly found in food, and its main sources are meat (turkey and chicken), liver, meat products, fish, nuts, whole grain products, milk, cheese, and eggs. Research into autoimmune bullous diseases has shown that nicotinamide in combination with minocycline is effective in the treatment of bullous pemphigus. These were observations carried out in a 76-year-old woman with comorbidities and a 46-year-old patient with esophageal involvement [46,47,55,56].

#### 3.2.2. Calcium, Potassium, Selenium, Zinc, and Copper

Calcium (Ca) is one of the basic building materials for bones and teeth. Additionally, it is involved in muscle contractility, conduction of nerve impulses, hormonal regulation, and activation of some enzymes. The best source of calcium is milk and its products. Significant amounts of this ingredient also contain products of plant origin, such as parsley leaves, kale, or spinach—however, calcium from these products is not as well absorbed compared to milk and its products. Calcium supplementation is recommended at the beginning of glucocorticosteroid treatment to prevent secondary osteoporosis. In the case of pemphigus treatment with chronic glucocorticosteroid therapy, supplementation of this mineral is recommended at a dose of 1000–1200 mg [57,58].

Potassium (K) is involved in regulating the osmotic pressure of cells, maintaining water and electrolyte balance, and the metabolism of proteins and carbohydrates. Large amounts of potassium are contained in nuts, seeds, dried fruits, chocolate, cocoa, vegetables, fruits, meat, and cereal products. Potassium supplementation is recommended in the treatment of pemphigus in people using chronic glucocorticoid therapy. The dose should be adjusted based on serum electrolyte levels routinely examined during glucocorticoid therapy [58].

Selenium (Se) takes part in the metabolic processes of the cell and protects against free radicals. Products rich in selenium include offal (especially kidneys), seafood (fish and crustaceans) and some vegetables (mushrooms, garlic, and dry legumes). In the human body, zinc (Zn) has structural, regulatory, and catalytic functions. It takes part in the metabolism of carbohydrates, fats, and proteins, and also influences the processes of memory and learning. It is found in liver, meat, brown bread, rennet cheese, eggs, and buckwheat. Copper (Cu) is a component of many enzymes, and it is involved in the metabolism of oxygen and the metabolism of iron in the body. Products rich in copper include wheat bran, offal (especially liver), sunflower seeds, nuts, and cocoa. It has been shown that patients with pemphigus vulgaris have lower serum concentrations of copper, selenium, and zinc compared to healthy volunteers. It is believed that these tendencies may be caused by improper nutrition associated with painful lesions in the oral cavity or chronic inflammation. Data on this association are limited and there are no clear guidelines regarding supplementation for these nutrient deficiencies in pemphigus [58,59].

### 3.3. Diets

According to the International Pemphigus and Pemphigoid Foundation (IPPF), there is no single diet that will help treat AIBDs. However, there are certain recommendations regarding products that exacerbate or alleviate the symptoms of the disease. Additionally, various types of diets have been described in the literature (including gluten-free, pulpy, or high-protein diets) that have a potential impact on the development or support the remission of the disease [60,61].

According to the IPPF, products that are bothersome for patients include the following: citrus fruits, sour fruits, bagels, garlic, chips, barbecue/cocktail sauces, horseradish, red pepper, onions, tomato sauces, chocolate, pickled cucumbers, tomatoes, roasted corn, pretzels, pizza, and coffee. It is recommended to rinse your mouth with water, hydrogen peroxide, or a moisturizing and soothing solution during and after eating to help remove food and bacteria and accelerate healing. In order to prevent malnutrition, it is recommended to eat a variety of foods every day, take vitamin supplements, and measure body weight weekly—and if weight loss occurs, increase the supply of kilocalories and protein. In the course of pemphigus and other autoimmune blistering diseases, diet plays an important role. It is classified as a treatment-supporting therapy, the main goals of which include maintaining the patient’s proper nutritional status, elimination of products and dishes that initiate or exacerbate the disease process, and introduction of products that have a positive impact on the course of the disease [61].

There are many factors that influence the nutrition (including nutritional status and diet) of patients with ABIDs. These include the following: (i) a number of dietary factors that are believed to play an important role in the onset, progression, exacerbation, and treatment of this disease; (ii) increased catabolism due to epidermal detachment and protein loss; (iii) hydroelectrolytic imbalance caused by fluid loss through skin lesions; and (iv) the need for vitamin D and calcium supplementation in the prevention of secondary osteoporosis as a result of long-term glucocorticoid therapy.

In patients who have difficulty swallowing or have complications after glucocorticosteroid therapy, it is recommended to establish an appropriate diet with a dietitian. The diets most commonly used in the treatment of pemphigus and other ABIDs are the soft diet, diet rich in protein, diet rich in calcium, DASH diet (Dietary Approaches to Stop Hypertension), Mediterranean diet, and gluten-free diet [46,62].

#### 3.3.1. Soft Diet

The pulp diet is similar to the easily digestible diet in terms of energy value and product selection but differs in consistency. It is recommended for diseases of the oral cavity and/or esophagus, for people who have problems with biting and/or swallowing. Its goal is to provide the patient with all the necessary nutrients and, as a result, prevent malnutrition. In pemphigus, due to the occurrence of blister-like changes with the formation of erosions on the mucous membranes of the mouth and/or throat, patients often have problems with biting and/or swallowing. This diet works well for these patients because the changed consistency makes it easier to eat. Dishes and products are served in a form that does not require biting. Very important ingredients are animal protein, B vitamins, vitamin C, and minerals. The main thermal treatment is cooking so as not to irritate the affected areas. Creamy and puree soups are widely used in this diet. Recommended products include crustless bread, soft, soaked in milk or tea, or crumbled; meats cooked or prepared from minced meat; and vegetables and fruits in raw form in the form of juices or cooked and crushed. It is recommended to eat four or five meals a day. The permitted spices are lemon balm, dill, cinnamon, marjoram, cloves, and vanilla. The assumptions for a pulpy diet include total protein (16% of the total dietary energy requirement, E), fats (30% E), and carbohydrates (54% E) [32,60].

#### 3.3.2. High-Protein Diet and High-Calcium Diet

The aim of a high-protein diet is to provide the appropriate amount of protein for the construction and reconstruction of body tissues. It is characterized by an increased supply of protein (15–20% E) with a limited intake of carbohydrates (below 40% E). When used for pemphigus, the protein supply is 2–3 g/kg body weight/day. Most (2/3) of this macronutrient should come from animal products, such as milk, cottage cheese, lean meats, and eggs. Patients should limit their intake of salt, salty snacks, and processed foods. A high protein supply in pemphigus is important due to the increased catabolism caused by epidermal detachment and potentially associated cancer. A high-protein diet is used in severe cases of pemphigus, as well as in the treatment of pemphigus with chronic glucocorticosteroid therapy. Its aim is to minimize protein loss and accelerate the healing of skin and mucosal lesions. In most patients, insertion of a nasogastric tube is required because erosions of the mucosa make oral feeding difficult. In patients able to eat independently, intermittent tube feeding is used, also at night. A nasogastric tube and possibly venous access are also necessary to correct fluid and electrolyte imbalances. Additionally, calcium and vitamin D supplementation is important to prevent secondary osteoporosis [60,62].

A high-calcium diet is characterized by an increased supply of calcium. Most of the intake of this mineral is found in milk and its products. It is mainly used for the osteoporosis that already accompanies pemphigus and in chronic glucocorticosteroid therapy. A calcium intake of 1000–1200 mg is recommended, which is also intended to prevent secondary osteoporosis associated with the use of glucocorticoids [63].

#### 3.3.3. The DASH Diet and the Mediterranean Diet

The DASH diet (Dietary Approaches to Stop Hypertension) has a beneficial effect on blood pressure and reduces the risk of coronary heart disease and stroke. It involves limiting the consumption of salt (less than 2.3 g of sodium/day), highly processed foods, sugar, and fats rich in saturated fatty acids; increasing the consumption of vegetables, fish, and fruit, as well as the supply of potassium, calcium, magnesium, and dietary fiber; and ensuring the right amount of protein. In autoimmune bullous diseases, this diet is used in chronic glucocorticoid therapy [60,64,65,66].

The Mediterranean diet is characterized by a low consumption of animal fats and a high consumption of plant products. In this diet, it is recommended to eat fresh fruit, bread, pasta, groats that are as minimally processed as possible, as well as whole-grain rice, yogurt, fish, low-fat dairy products, and nuts. The consumption of meat, eggs, yellow and fromage cheeses, and sugar is limited. Similarly to the DASH diet, it is used in autoimmune bullous diseases in which glucocorticoids are used chronically [67,68,69].

#### 3.3.4. Gluten-Free Diet

In dermatitis herpetiformis, it is advisable to follow a gluten-free diet and eliminate iodine from the diet (fish and seafood, iodized table salt, sea algae, mineral water containing iodine) [70]. Well-controlled patients with Duhring’s disease treated by a dermatologist, gastroenterologist, and dietitian have an excellent prognosis. When recommending a diet low in iodine, remember that it is necessary for the body’s metabolic processes, especially those regulated by thyroid hormones, and is very important for the proper growth and development of children. The mechanism by which iodine compounds intensify skin lesions is not sufficiently understood. Patients should avoid staying in coastal areas where there is a high content of iodine compounds. Taking mineral and vitamin preparations containing iodine compounds may also worsen the symptoms of the disease. Patients with dermatitis herpetiformis are advised to avoid all products containing gluten (which is a mixture of proteins: gliadin, gluten, secalin, avenin, and hordein), and made from cereals, including wheat (also spelt, emmer, einkorn, and durum wheat), barley, rye and malt. Gluten may also be found in food products from other groups to which ingredients derived from gluten-containing cereals in any form have been added (e.g., wheat fiber, wheat starch syrup, pure gluten, breadcrumbs, etc.). These ingredients can be added to popular cold meats (lunch meat and sausages), delicatessen products, some sweets, dairy products, food concentrates, spices, canned goods, ready-made sauces, desserts, etc. When buying ready-made products, always check their composition on the labels. Gluten-free products are marked with a certificate number and often with the Crossed Grain symbol. This mark on certified gluten-free products is an international symbol of safe gluten-free food. Certification means that the product is made from safe raw materials and regularly tested for gluten content [60,71,72,73]. It is worth noting that DH patients can consume pure oat products. Recent studies have shown that oats are not only safe, but in the long run they can improve patient quality of life and eliminate any gastrointestinal symptoms. However, patients should exercise caution because most store-bought oat products are usually contaminated with gluten, so it is recommended to avoid oats or products containing this grain. A good solution is to buy certified oat products. Examples of gluten-free foods that are safe to eat include rice, corn, potatoes, vegetables, and others listed in Table 3. The table also includes products that are prohibited in a gluten-free diet. When preparing meals with gluten-free products at home, be careful not to cross-contaminate them with gluten, e.g., by using the same cutting board for cutting gluten-free and regular bread or the same spoon for gluten-free sauce and that thickened with wheat flour [60].

Adherence, unfortunately, is difficult for patients in practice because it requires meticulous monitoring of food labels and consumption, can be costly and inconvenient, and is socially limiting. Studies from many centers confirm that compliance with the diet depends to the greatest extent on patients’ knowledge about the disease and its treatment. Dietitians and support groups are helpful in dealing with compliance challenges and pinpointing hidden sources of gluten. Patients can also benefit from the support of various associations, e.g., the Polish Association of People with Celiac Disease and on a Gluten-Free Diet (https://celiakia.pl (accessed on 8 August 2024)) [74]. Communication with the patient and family should emphasize the importance of following an appropriate diet even in the absence of symptoms.

Over the past two decades, several studies have assessed the possibility of long-term remission of Duhring’s disease in 10–20% of cases, which has suggested the possibility of discontinuing the gluten-free diet in well-controlled patients. However, it was recently shown that 95% of patients who were well controlled on long-term gluten-free diet therapy relapsed after gluten challenge. Therefore, for now, it is recommended to use this diet throughout life. It is also recommended to constantly monitor the patient for compliance with dietary recommendations and the risk of nutritional deficiencies resulting from the use of the elimination diet [10].

A gluten-free diet affects both skin and intestinal changes. Please remember that its effects are only noticeable after many months of use (minimum 6 months). Its scrupulous use can even protect many patients from the use of pharmacotherapy. It is worth emphasizing the observation that the use of a gluten-free diet in patients with dermatitis herpetiformis also reduces the risk of developing gastrointestinal lymphomas [71].

### 3.4. Herbal Supplements

The use of herbal supplements, which can improve the functioning of the immune system, has become popular among patients with dermatological diseases. However, recommending any herbal preparations to patients with autoimmune bullous diseases should be done very cautiously, as herbal preparations may worsen the disease. The herbs that enhance the disease process in pemphigus include algae (*Spirulina platensis*), *Echinacea*, and St. John’s wort (*Hyperitum perforatum*). Algae (*Spirulina platensis*), which have been called “super foods” due to their high protein content and their possible hypolipidemic, antioxidant and anti-inflammatory effects, have a probable impact on the induction of pemphigus vulgaris. Echinacea, species of plant from the Asteraceae family which comes in several varieties, the most popular of which are purple coneflower and *Echinacea angustifolia*, enhances the body’s response to inflammation and infection, and has a probable influence on the induction of pemphigus vulgaris. St. John’s wort (*Hyperitum perforatum*), a herbal medicine with a widely known photosensitizing effect and possible immunomodulatory effect, has a probable influence on the induction of pemphigus foliaceus [75,76].

However, *Cassia fistula* seems to have some use in the treatment of pemphigus. It is a tropical, deciduous, green tree from India, with an upright and slender trunk and small buttresses. *C. fistula* fruit oil contains lupeol and anthraquinone compounds such as rhein and flavonoids. It is believed that this herb may be a recommended botanical therapeutic support in the treatment of erosions in pemphigus vulgaris [17].

### 3.5. Fast Foods

Fast food is a type of highly processed food, prepared quickly and eaten hot, and usually cheap. These include products such as pizza, hamburgers, fries, etc. These foods may influence the development of pemphigus. The pathomechanism is not fully known and requires further research [2]. However, there are several factors to consider regarding how fast foods might affect these conditions, including their nutritional quality, impact on gut health, weight gain, and obesity, and specific ingredients. Diets rich in saturated fats and sugars can trigger inflammation, potentially worsening autoimmune conditions, including AIBDs. A deficiency in essential nutrients that happens when consuming fast foods, such as omega-3 fatty acids, vitamins A, C, and D, and zinc, can impair immune function and skin health, which may aggravate the symptoms of AIBDs. Moreover, a diet rich in fast foods can alter the balance of gut microbiota, resulting in dysbiosis. This imbalance may affect systemic inflammation and immune responses, potentially triggering or worsening autoimmune reactions. Obesity is associated with increased levels of inflammatory cytokines, which can exacerbate autoimmune conditions, including AIBDs. Finally, certain fast foods include additives, preservatives, and artificial ingredients that can provoke inflammatory responses or allergic reactions in sensitive individuals. For individuals with conditions such as dermatitis herpetiformis, which is associated with gluten sensitivity, fast food items that contain gluten can worsen skin symptoms [77,78,79].

## 4. Dietary Management in Autoimmune Bullous Diseases and the Role of Dieticians

### 4.1. Dietary Management in Autoimmune Bullous Diseases Treated with Chronic Glucocorticosteroid Therapy

The advent of corticosteroids in the early 1950s revolutionized the prognosis for many autoimmune bullous diseases and led to a drastic decline in patient mortality. Glucocorticosteroids (GCSs) are the most commonly used drugs in skin diseases. Many patients with autoimmune bullous diseases use glucocorticosteroid therapy chronically, i.e., for more than 3 months. The most commonly used glucocorticoids include prednisone, prednisolone, and dexamethasone [80].

There are many beneficial aspects of using glucocorticoids in the treatment of AIBDs, the most important of which is the reduction of mortality among these patients. However, GKSs also have the negative effects of impaired calcium absorption from the gastrointestinal tract, increased loss of calcium in urine, and inhibition of the synthesis of 25-hydroxycholecalciferol in the liver and 1,25-dihydroxycholecalciferol in the kidneys, which ultimately leads to osteopenia and/or osteoporosis [81].

However, it must be emphasized that the choice between life and side effects that can be lived with should not be shifted towards increasing the risk of death. Osteoporosis is a chronic, progressive disease manifested by low bone mass and degradation of the microarchitecture of bone tissue. This disease leads to weakening of bone strength, which consequently increases the risk of fractures. Osteopenia is the early stage of osteoporosis. Studies indicate a high percentage of patients with osteopenia and osteoporosis in pemphigus treated with GCSs. According to one of them, this percentage was as high as 41.9%. It is estimated that significant bone loss and bone fractures affect 30–50% of patients with pemphigus. Within 3–6 months of therapy, this risk increases dramatically. Moreover, patients often struggle with cardiovascular diseases and insulin resistance. Therefore, in the case of long-term glucocorticosteroid therapy, in addition to lifestyle modeling related to stopping smoking and drinking alcohol, it is recommended (depending on the complications related to glucocorticosteroid therapy) patients maintain a diet: high-protein, high-calcium, DASH, Mediterranean, or a combination of these. It is also recommended to supplement calcium, vitamin D, and potassium. The general scheme of dietary management when using chronic glucocorticoid therapy in autoimmune bullous diseases is presented in Figure 1. It is important to outline that under no circumstances should dietary therapy be used to modify the dose of glucocorticoids taken by patients with autoimmune bullous diseases. Such action can only increase the mortality rate of patients [82,83].

### 4.2. Dietary Management and the Risk of Cardiovascular Diseases in Autoimmune Bullous Diseases

Recently, there has been a lot of interest in the relationship between diet and the risk of developing cardiovascular diseases in patients with pemphigus. This is probably due to their high frequency in this group of patients. According to one of the first studies in this area, from 2022, which assessed the impact of dietary diversity on the risk of cardiovascular diseases among 187 patients aged 18–65 with pemphigus vulgaris, no connection was clearly demonstrated between the above-mentioned factors. There was no relationship between the dietary diversity index and the occurrence of obesity and glucose homeostasis disorders. However, the results of this cross-sectional study showed that the dietary diversity index may be associated with increased concentrations of total cholesterol and HDL (*high-density lipoprotein*) cholesterol. However, further prospective studies are needed to confirm these observations [82]. Moreover, according to a randomized, double-blind study, the use of L-carnitine supplements in patients with pemphigus vulgaris had a beneficial effect on cystatin C, which led to favorable changes in markers of cardiovascular health and bone turnover [83].

### 4.3. The Role of Dieticians in Autoimmune Bullous Diseases

A dietitian plays an important role in the interdisciplinary team caring for a patient with autoimmune bullous disease. According to the consensus of the Polish Society of Dermatology from 2014, dietary consultation is considered a supportive treatment for pemphigus and other autoimmune skin diseases, especially dermatitis herpetiformis. It is especially recommended for patients with swallowing difficulties and complications after glucocorticosteroid therapy. The main tasks of a dietitian include the following: conducting a nutritional interview with the patient and assessing their nutritional status (such as anthropometric tests); learning about the individual varieties and clinical subtypes of pemphigus and the risk factors associated with them in order to include an appropriate diet; locating disease symptoms and their spread (e.g., skin lesions and erosions in the oral cavity); patient education regarding nutritional treatment for their disease; own education on autoimmune bullous diseases, including pemphigus and especially dermatitis herpetiformis (Duhring’s disease); and monitoring the patient’s health condition [18,84].

The aim of the dietary consultation is to ensure the patient’s appropriate nutritional status (including the prevention of malnutrition), maintain appropriate body weight by implementing appropriate nutritional treatment, and consequently improve the quality of life of patients. When determining your diet, you should take into account the patient’s current weight and height (determination of body mass index, BMI), medications they take, current comorbidities, patient’s physical activity, patient’s dietary preferences, and permitted and prohibited products [85].

Despite the lack of clear guidelines regarding diet in AIBDs, especially pemphigus, based on the available literature, general dietary recommendations can be formulated for patients regardless of the type of this disease [60]. These are lifestyle modifications by eliminating the use of stimulants (alcohol and cigarettes) and increasing physical activity; limiting or eliminating the consumption of products and foods rich in compounds that are known to cause or worsen the symptoms of pemphigus, such as thiols, phenols, tannins, etc., which are often found in foods, drugs, and cosmetic products; establishing an appropriate diet with a dietitian depending on the patient’s health condition, clinical symptoms, and risk factors related to the use of chronic glucocorticosteroid therapy (particular importance is attached to a gluten-free diet in dermatitis herpetiformis and a diet used in swallowing disorders in other autoimmune bullous diseases); appropriate supplementation of vitamins, minerals, and L-carnitine (especially in the case of chronic glucocorticosteroid therapy); drawing attention to the usefulness and side effects of herbal supplements for patients with pemphigus and other autoimmune blistering diseases (it is important, however, not to use them as replacements for treatment—the educational role of a dietitian is crucial); monitoring the health condition as well as providing the patient with constant dietary control (especially for patients with severe pemphigus); as well as taking an interdisciplinary approach to treatment [86,87].

### 4.4. Support Groups

In Poland, there are currently no support groups for pemphigus patients and their families. Patients should be encouraged to create them and cooperate with groups operating in the USA and Western Europe, such as the Pemphigus Vulgaris Network (www.pemphigus.org.uk (accessed on 20 July 2024)), Pemphigus-Pemphigoid-France (www.pemphigus.asso.fr (accessed on 20 July 2024)), and the International Pemphigus and Pemphigoid Foundation (www.pemphigus.org (accessed on 20 July 2024)) [61,88,89].

Patients with Duhring’s disease can be advised to contact and read the information available on the website of the Polish Association of People with Celiac Disease and the Gluten-Free Diet (www.celiakia.pl (accessed on 8 August 2024)) [79].

## 5. Discussion

This review aimed at summarizing the role of nutrition in pathogenesis and treatment of the following AIBDs: (i) pemphigus, (ii) bullous pemphigoid and mucous membrane pemphigoid, (iii) dermatitis herpetiformis, and (iv) epidermolysis bullosa acquisita. Several studies have been found on this topic.

The current evidence suggests that the relationship between food and AIBDs is complex [26]. There is some evidence confirming the influence of nutritional factors and mineral deficiencies on the development of pemphigus [27,28,29,30]. In one analysis, 46.2% of pemphigus patients mentioned food as a possible trigger for their disease [30]. Dietary factors that exacerbate, precipitate, and/or relapse the disease include the following: thiols and onion vegetables (*Allium*) (such as garlic, onion, leek, and chives); phenols (mango, pistachio, red pepper, black pepper, cinnamon, and fennel); tannins (coffee, tea, raspberry, guarana, cranberry, avocado, and wild strawberry); phycocyanin (cryptophytes, red algae, and cyanobacteria); isothiocyanate (white and black mustard, horseradish, and nasturtium); all trans-retinoic acids; cinnamic acid (tomato, orange, and grapefruit); walnut antigens; fast food; and herbs (algae—*Spirulina platensis*, horsetail herb, rosehip, echinacea, and ginseng). Many patients with AIBDs may require supplementation, particularly of vitamins D and B_3_, calcium, potassium, zinc, selenium, and cobalt. A soft diet is recommended for individuals with oral cavity or esophageal issues, especially for those who have difficulty biting or swallowing, and is commonly used in managing pemphigus. For long-term glucocorticoid therapy, a high-protein, high-calcium diet, along with the DASH (Dietary Approaches to Stop Hypertension) diet and Mediterranean diet, is advised. In cases of dermatitis herpetiformis, a gluten-free diet with iodine elimination is recommended. Algae (*Spirulina platensis*), Echinacea, and St. John’s Wort (*Hypericum perforatum*) may exacerbate autoimmune blistering diseases, while *Cassia fistula* could be beneficial in treating erosions in pemphigus vulgaris. Fast foods may contribute to the development of autoimmune blistering diseases, however, the underlying pathomechanism remains unclear. While these nutrients can have implications for individuals with AIBDs, it is important for patients to consult with healthcare professionals, such as dietitians or allergists, to tailor dietary approaches based on their specific conditions and sensitivities [90,91].

Besides several strengths that this review contains, such as its comprehensive scope, identification of dietary triggers, as well as discussion on nutrients and supplementation needs, this study contains also several limitations. Most of the studies presented in this review focused on the negative, rather than protective, impacts, especially regarding nutrients. This is the limitation which represents a negative bias that can influence both the design and interpretation of studies in the following ways: study selection bias, reporting bias, and publication bias. In this context, this may distort perceptions, making negative effects seem more common or significant than they would if positive outcomes were equally highlighted.

Future researchers should prioritize high-quality research on this topic. These include, e.g., longitudinal studies examining the relationship between dietary habits and disease activity in patients with autoimmune bullous diseases, establishing clearer connections between diet and disease management, or controlled dietary interventional studies which investigate the impact of specific diets, such as gluten-free, high-protein, and Mediterranean diets, on clinical outcomes in patients with AIBDs, focusing on symptom relief and disease progression [92,93]. The current literature emphasizes dietary factors that exacerbate AIBDs, with less focus on protective or therapeutic dietary components. To address this gap, future research could explore nutrients or food compounds with potential anti-inflammatory or immune-modulating properties. For instance, omega-3 fatty acids—found in sources like fish oil and flaxseed—are known to reduce inflammation in other autoimmune diseases and may be worth investigating for their potential benefits in AIBDs. Furthermore, due to the variability in how patients respond to dietary factors, precision nutrition—customizing dietary guidance based on individual genetic, microbiome, and metabolic profiles—could be highly advantageous. What is more, future research could investigate how genetic predispositions and microbiome differences impact dietary triggers or protective elements in AIBDs, paving the way for more personalized dietary recommendations. Although the review identifies fast food as a potential contributor to AIBDs, it would be helpful to clarify which specific components (such as preservatives, high-fat content, and additives) may influence autoimmune responses. Future research could investigate the effects of particular food additives—like artificial colors, preservatives, and emulsifiers—on AIBDs symptoms and disease pathology. Finally, psychological factors, including stress, are known to affect both autoimmune disease progression and dietary choices. Future research could examine the impact of combining stress-reduction strategies with dietary adjustments to offer a more holistic approach to managing these diseases. Additionally, socioeconomic factors may influence diet quality and access to nutritious foods, making them essential considerations in the broader management of AIBDs at the population level.

## 6. Conclusions

Research on the use of a gluten-free diet in dermatitis herpetiformis is confirmed in the literature beyond any doubt. Research on the effects of garlic and the thiols and phenols that may cause pemphigus is well established, but there are many other ingredients that do not have enough scientific support to recommend avoiding them. Nevertheless, based on the available literature, general dietary recommendations for patients and guidelines for dietitians can be formulated to support the treatment process, which can be particularly fruitful if an interdisciplinary approach is used. However, further research is needed to more precisely define the relationships between nutrients and nutrition and blistering diseases.

## Figures and Tables

**Figure 1 nutrients-16-03961-f001:**
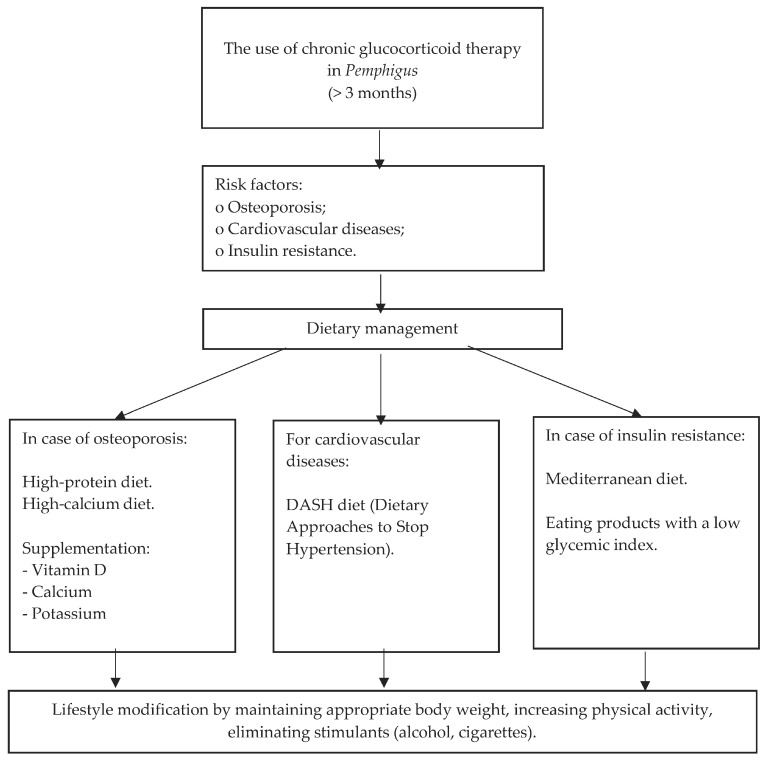
Scheme of dietary management when using chronic glucocorticoid therapy in *Pemphigus*.

**Table 1 nutrients-16-03961-t001:** Classification and characteristics of the Autoimmune Bullous Diseases.

	Definition	Incidence [per year]	Persons Affected	Diagnosis	Treatment
Pemphigus	pemphigus is an autoimmune disease, potentially life-threatening, in which the immune system is dysregulated, and autoantibodies are formed against desmogleins, i.e., surface proteins of keratinocytes, namely desmoglein 3 and/or 1	0.5–8/ 1,000,000	all age groups with the peak incidence around the age of 50–60, women are affected more often	clinical picture of the lesions and the results of immunological tests optionally histopathological examination of skin and/or mucous membrane sections	glucocorticoids combined with immunosuppressive Drugs, rituximab
Bullous pemphigoid and mucous membrane pemphigoid	bullous pemphigoid is the most common autoimmune bullous disease caused by autoimmune reaction against antigen BP 180; typical lesions are subepidermal tense blisters, on erythema and urticarial skin and on normal looking skin	0.2–4/ 100,000	elderly people over 60 years of age, with no gender predilection	diagnose is based on the clinical picture and immunopathological test results, optionally histopathological examination	glucocorticoids (topically or systemic); methotrexate, tetracyclines, azathioprine, mycophenolate mofetil, cyclophosphamide and dapsone
Dermatitis herpetiformis (During’s Diseases)	is a chronic, intensely itchy skin autoimmune blistering disorder connected with gluten-sensitive enteropathy characterized by clusters of small blisters and red, raised patches, typically appearing on the elbows, knees, buttocks, and scalp	0.4–3.5/ 100,000	slightly more often affecting men than women, 4th and 5th decade of life	clinical picture of the lesions, and immunological tests and possibly by histopathological examination	gluten free diet and in some cases small doses of dapsone
Epidermolysis Bullosa Aqusita	an autoimmune subepidermal bullous disease in which the autoimmunity process is directed against epitopes of collagen VII, which is a protein that forms anchoring fibers connecting the basement membrane with the dermis.	0.25/ 1,000,000	children and adults of all ages	clinical picture and immunological tests and possibly by histopathological examination	glucocorticoids, sometimes in combination with sulfones. Other options include colchicine and in severe cases immunosuppressive drugs or immunoglobulin infusions

**Table 2 nutrients-16-03961-t002:** The Role of Nutrition in the Pathogenesis and Treatment of Autoimmune Bullous Diseases.

Name	Source	The Role in Pathogenesis and Treatment of AIBDs
**1. Nutrients**		
Thiols and bulb vegetables (*Allium*)	garlic, onion, leek, chives	15% of patients reported garlic as a trigger for pemphigus; many patients declared that their symptoms were exacerbated and/or triggered after consuming leeks and garlic, and they declared that their symptoms disappeared after eliminating garlic, onions and leeks from their diet.
Phenols	pistachio, cinnamon, mango, red pepper, black pepper, thymol, eugenol, aspartame, cinnamon bark oil, pea seeds, rosmarinic acid, cynarin (artichoke), allspice, fennel, curcumin, arbutin, mace, milk	involved in the induction of pemphigus in genetically predisposed patients; elimination of phenolic compounds from the diet absolutely does not allow any reduction in the dose of systemic glucocorticoids in these patients.
Tannic Acid	coffee, tea, eggplant, cassava, cherries, blackberries (leaves), cranberries, ginger, avocados, oak bark, blueberries, wild strawberries, sage, willow bark, walnuts, cashew nuts, rosemary, ground pepper (betel), cassava and mango	tannins play a special role in the case of endemic pemphigus; these compounds are involved in the induction of pemphigus, especially in genetically predisposed patients; eliminating tannins from the diet absolutely does not allow for reducing the dose of systemic glucocorticoids in these patients.
Tannins	tree bark, plants, black pepper, cherries, blueberries, mangoes, cashews, tea, vanillin and cocoa	Research shows that patients who consume large amounts of tannins in their diet also have higher concentrations of them in their skin. These communities are suspected to be disproportionately affected by the disease due to their proximity to rivers with high tannins in their water systems and diet. Additionally, the diagnosis rate of pemphigus vulgaris is high in India, which may be related to the consumption of large amounts of tea and betel nut.
Phycocyanin and Isosulfurcyanates	phycocyanin is a blue pigment found in cryptophytes, cyanobacteria and red algae. Isosulfurcyanates are found in mustard.	phycocyanin is responsible for the induction of pemphigus, especially in genetically predisposed patients; the elimination of phycocyanin from the diet does not allow for any modification of the dose of systemic glucocorticoids in these patients. isosulfurcyanates are responsible for the induction of pemphigus, especially in genetically predisposed patients; eliminating isosulfurcyanates from the diet does not allow for reducing the dose of systemic glucocorticoids in patients.
All trans-retinoic acids and Cinnamic Acid	cinnamic acid is found in candied fruit, tomatoes, oranges and grapefruits	all trans-retinoic acids shift the balance of the Th1 to Th2 ratio towards Th2, which is known to be elevated in patients with pemphigus; In one study in which pemphigus patients were asked to list possible triggers, the most frequently mentioned food-related trigger was tomato (23.1%), which is high in cinnamic acid.
Walnut Antigens	walnut antigens	Research suggests that exposure to walnut antigen through gastrointestinal epithelial cells may activate B cells in individuals genetically predisposed to pemphigus vulgaris.
**2. Vitamins and Minerals**		
Vitamin D	endogenous synthesis (approx. 80%), products of animal origin (oils, fatty fish, eggs)	it was found that patients with pemphigus vulgaris had lower vitamin D levels; its supplementation is important in the case of pemphigus treatment with the use of chronic glucocorticosteroids therapy and during long-term treatment of pemphigus with the use of immunosuppressive drugs.
Vitamin B3	meat (turkey, chicken), liver, meat products, fish, nuts, whole grain products, milk, cheese and eggs	Nicotinamide in combination with minocycline is effective in the treatment of bullous pemphigus.
Calcium	milk and its products, parsley leaves, kale or spinach	Calcium supplementation is recommended at the beginning of glucocorticosteroid treatment to prevent secondary osteoporosis.
Potassium	nuts, seeds, dried fruits, chocolate, cocoa, vegetables, fruits, meat and cereal products	Potassium supplementation is recommended in the treatment of pemphigus in people using chronic glucocorticoid therapy.
Selenium	offal (especially kidneys), seafood (fish and crustaceans) and some vegetables (mushrooms, garlic, dry legumes)	It has been shown that patients with pemphigus vulgaris have lower serum concentrations of copper, selenium and zinc. It is believed that these tendencies may be caused by improper nutrition associated with painful lesions in the oral cavity or chronic inflammation.
Zinc	liver, meat, brown bread, rennet cheese, eggs and buckwheat	
Cobalt	wheat bran, offal (especially liver), sunflower seeds, nuts, and cocoa	
**Name**	**The Role in Pathogenesis and Treatment of AIBDs**	
**3. Diets**		
Soft diet	It is recommended for diseases of the oral cavity and/or esophagus, for people who have problems with biting and/or swallowing. Its goal is to provide the patient with all the necessary nutrients and, as a result, prevent malnutrition. In pemphigus, due to the occurrence of blister-like changes with the formation of erosions on the mucous membranes of the mouth and/or throat, patients often have problems with biting and/or swallowing.	
High-protein and high- calcium diet	It is mainly used in the osteoporosis that already accompanies pemphigus and in chronic glucocorticosteroids therapy.	
DASH diet	In autoimmune bullous diseases, this diet is used in chronic glucocorticoid therapy.	
The Mediterranean diet	Similarly, to the DASH diet, it is used in autoimmune bullous diseases in which glucocorticoids are used chronically.	
A gluten-free diet	It is advisable to follow a gluten-free diet and eliminate iodine from the diet (fish and seafood, iodized table salt, sea algae, mineral water containing iodine) in dermatitis herpetiformis.	
**4. Herbal supplements**		
Algae (*Spirulina platensis*)	All of them enhances the ABIDs. *Hyperitum perforatum* has a probable influence on the induction of pemphigus foliaceus.	
*Echinacea*		
St. John’s wort (*Hyperitum perforatum*).		
*Cassia fistula*	It is believed that this herb may be a recommended botanical therapeutic support in the treatment of erosions in pemphigus vulgaris	
**5. Fast-foods**		
pizza, hamburgers, fries, etc.	These foods may influence the development of pemphigus. However, the pathomechanism is not fully known and requires further research.	

**Table 3 nutrients-16-03961-t003:** Cereals, seeds and other sources of starch allowed and prohibited in a gluten-free diet [72].

Naturally Gluten-Free Products—Cereals, Seeds and Other Sources of Starch	Cereals and Their Derivatives Containing Gluten
Amaranth (amaranth)	Wheat (including spelled, kamut, emmer, einkorn, durum wheat)
Ararut (arrowroot)	
Sweet potato (sweet potato, yam)	
Carob (carob, locust bean flour)	Barley
Chia (Spanish sage)	
Fonio (digitaria, fingerstick)	Rye
Buckwheat and buckwheat groats	
Indian rice grass (montina)	Triticale (triticale)
Cocoa	
Koiks of Job’s tears (tearism)	Groats: bulgur, Kraków, couscous, semolina, Masurian, pearl, barley, rural, Kujawy
Coconut	
Konjac	
Corn (maize, teosinie), corn flour, corn flakes	Wheat sprouts
Cassava (cassava, tapioca, yucca)	
Seeds and seeds: pumpkin, poppy, sunflower	
Nuts: Brazil, chestnuts, hazelnuts, macadamia, almonds, cashews, pecans, pistachios, pine nuts, walnuts, acorns	Durum flour (durum, semolina), rye, barley, emmer, einkorn, spelled
Oat *	
Millet (ragi, millet)	Seitan
Quinoa (quinoa, Peruvian rice)	
Rice (brown, red, wild, white)	Barley malt
Sago	
Sesame	Wheat, rye, barley starch
Linseed	
Sorghum (milo)	Wheat germ oil
Legumes: broad beans, chickpeas, peas, mesquite, lupine, peanuts, lentils, soybeans	Wheat, rye and barley bran and flakes
Tara	
Tef/Teff (Abyssinian grass, Abyssinian grass) Jerusalem artichoke (bush sunflower) Potato	Wheat germ

* oats are a gluten-free cereal, but may be contaminated during cultivation, harvesting and processing.

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
