# Peer review of "The Role of Nutrition in the Pathogenesis and Treatment of Autoimmune Bullous Diseases—A Narrative Review"

_nutrients, 2024, doi:10.3390/nu16223961_

Round 1

Reviewer 1 Report (New Reviewer)

Comments and Suggestions for Authors

The current manuscript is limited to presenting qualitative and abstract content on the effects of foods. This may lead to the error of simplistic generalization.

Abstract: The key findings of this review need to be presented. The current abstract does not capture the comprehensive results of the review at all. Presenting the key conclusions will greatly help to appeal the importance of the review contents to the readers.

Lines of 70-72: It is strongly recommended that illustrations be provided to visually compare the differences between the four types of ABIDs classified by the authors (i) pemphigus, ii) bullous 70 pemphigoid and mucous membrane pemphigoid, iii) dermatitis herpetiformis, and iv) 71 epidermolysis bullosa aqusita). This will be very useful for readers who lack deep expertise in ABIDs to understand the types of diseases. The simple descriptive text in this manuscript does not allow for easy comparison and understanding of the four types of ABIDs classified by the authors.

Table 1: The information presented in the table is incomplete. There is no consistency between the contents presented in the table caption and the table body. In other words, information on the role of nutrition and foods in AIBDs is not provided at all. Additional classification information for columns or rows in the table is needed.

Lines of 250-251: “Tannins” is duplicated.

Figure 1: This is not a drawing. It does not follow the typical form of a schematic diagram. It is not clear what information the drawing is intended to convey.

Lines of 677-679: The expression needs to be clearer and more specific. Does this mean that a gluten-free diet is significantly helpful in alleviating dermatitis?

Lines of 679-680: “Research on the effects of garlic and the thiols and phenols that may cause pemphigus is well established” Please clearly present information on the limit doses that affect skin toxicity.

Table 2: The caption and the contents of the table body do not match. What information exactly is Table 2 intended to convey? It is difficult to grasp its importance.

The current manuscript is limited to presenting qualitative and abstract information on the effects of foods. Quantitative information needs to be clearly presented. Information such as the approximate maximum non-toxic dose for each food or the dose required to cause toxic reactions must be provided. This will greatly help prevent readers from misunderstanding the qualitative and abstract information presented in this manuscript.

It is strongly recommended to present quantitative information related to skin reactions for each food in a summary table.

Lines of 684-686: The content is very abstract. “High quality research” It is necessary to clearly state what specific research is needed.

Comments on the Quality of English Language

Repeated identical expressions in a manuscript make the manuscript boring.

Author Response

Dear Sir/Madame,

thank you for all your valuable comments and suggestions. 

Please find our answers in the attachment.

Kind regards,

AK

Reviewer 2 Report (New Reviewer)

Comments and Suggestions for Authors

The paper describes the role of nutrition in the development and management of autoimmune bullous diseases which are characterized by the formation of blisters and erosions on the skin and mucous membranes. The paper addresses the role of nutrition in the pathogenesis and treatment of the main AIBDs: pemphigus, bullous pemphigoid and mucous membrane pemphigoid, dermatitis herpetiformis, and epidermolysis bullosa aquisita.

In my opinion, the paper addresses all the main issues in an adequate manner, is written in an academic style yet easy to understand. The main nutrients are presented and their role described in detail. Additionally, the authors formulated their own conclusions on the topic.

I do however consider that the role of fast food in the general picture of autoimmune bullous diseases was very poorly addressed particularly considering the vast number of people that consume such foods on a daily basis. I suggest the authors to introduce new information on the topic, including the presentation of already reported theories.

Author Response

Dear Sir/Madame,

thank you for all your valuable comments and suggestions. 

Please find our answers in the attachment.

Kind regards,

AK

Reviewer 3 Report (New Reviewer)

Comments and Suggestions for Authors

This article discusses the complex relationship between autoimmune bullous diseases (AIBDs) and nutrition, and various foods and food ingredients can serve as disease regulators. The article summarizes the role of nutrition in the onset and treatment of some autoimmune bullous diseases, with a focus on the impact of dietary adjustments, various diets, and the nutritional consequences of these diseases. However, this article still has some flaws:

1. Why is it " 2. Characteristics of Selected Autoimmune Bullous Diseases " instead of "2 Characteristics of autoimmune bullous disease?

2. In "3", since the relationship between nutrition and AIBD is multifaceted, harmful and beneficial should be separately categorized under subheadings;

3. The full text lacks figures;

4.The full text lacks a discussion section;

5. "3.1.1 Thiols and bulb vegetables (onions)", it is not appropriate to put chemicals and food together;

6. Table 1 and Table 2 should be displayed at the specific content of the article;

7. Sections 3.1, 3.2, and 3.3 have incorrect numbering;

8. Some abbreviations in the text do not have their full names written when they first appear, so a list of abbreviations can be added;

9. The reference format is not uniform, and reviews generally require around 100 references.

Author Response

Dear Sir/Madame,

thank you for all your valuable comments and suggestions. 

Please find our answers in the attachment.

Kind regards,

AK

Round 2

Reviewer 3 Report (New Reviewer)

Comments and Suggestions for Authors

The author answered all my questions and made revisions to most of them. The author provided reasonable explanations for those that were not modified. The revised manuscript has made significant improvements. I am satisfied with this.

This manuscript is a resubmission of an earlier submission. The following is a list of the peer review reports and author responses from that submission.

Round 1

Reviewer 1 Report

Comments and Suggestions for Authors

Some points need to be addressed in a revised version of manuscript.

Pemphigus. Lines 54-

In addition to  pemphigus vulgaris, pemphigus foliaceus and paraneoplastic pemphigus there is also immunoglobulin A (IgA) pemphigus.

Please consider that multiple factors may be contributing to the increased incidence of blistering disease, particularly pemphigoid,including the aging population and associated increase in neurologic conditions (which may be pathogenically linked), increased use of drugs implicated in drug-induced disease, and an increase in the diagnosis of variant pemphigoid forms including localized bullous pemphigoid

Lines 214-5

There is a lot of scientific evidence confirming the influence of nutritional factors and  mineral deficiencies on the development of pemphigus.

Actually the cited references can at most suggest the possible role of diet in the worsening of pemphigus.

According  to the latest scientific research, 15% of patients reported garlic as a trigger for pemphigus. Actually the latest scientific research refers to the repetition of the same reference (26 and 28), dated 2003, which mentioned the toxic effect of topical application of phenol, while the other two references concerned the effect on IL-1 and a survey respectively. I question the importance of these references in supporting the Authors’ statement.

Paragrapf 3.1.3

Authors should report the interesting observation about endemic pemphigus.

 Endemic pemphigus foliaceus (fogo selvagem) has been reported in the region in the state of Mato Grosso do Sul in Brazil, where the prevalence of pemphigus foliaceus during the mid-1990s was around 3 percent [J Invest Dermatol. 1996 Jul;107(1):68-75. ]. An endemic form of pemphigus vulgaris also has been reported in a small number of patients residing in an endemic region of pemphigus foliaceus in Brazil [Arch Dermatol. 2007 Jul;143(7):895-9.

3.3.6

In dermatitis herpetiformis, it is advisable to eliminate  iodine from the diet (fish and seafood, iodized table salt, sea algae, mineral water containing iodine).

This statement is not supported by scientific evidence!

Comments on the Quality of English Language

English is fine (minor editing)

Author Response

Dear Sir/Madame, 

thank you for all your valuable comments. We have revised the manuscript accordingly. Please find our responses in the attachment. We would greatly appreciate any further suggestions you may have for improving our manuscript.

Reviewer 2 Report

Comments and Suggestions for Authors

In this manuscript, the roles of nutrition in pathogenesis and treatment of the autoimmune bullous diseases (AIBDs), including pemphigus, bullous pemphigoid and mucous membrane pemphigoid, dermatitis herpetiformis, and epidermolysis bullosa aqusita, were summarized. Generally, it is an interesting topic. The manuscript needs to be improved before it may be accepted for publication, especially for the introduction section and providing some related figures. Followings are some suggestions for revisions.

1. More information can be provided in the abstract.

2. It is a very short introduction section. Please introduce more background information. In the last paragraph please emphasized the key innovations of present review.

3. The phrases “Epidemiology and etiopathogenesis”, “Clinical picture”, and “Diagnosis and pharmacological treatment” in the section of “2. Characteristics of Selected Autoimmune Bullous Diseases” can be deleted and describe them in the first sentence of related paragraph, respectively.

4. Please avoid using the first-person “we” narratives for a scientific paper.

5. The tables should be provided as three-line table.

6. Some related figures can be provided to vivid illustrate the results.

7. The Latin names of plants, such as “Spirulina platensis”, should be in Italics font style.

8. The “3.2. Vitamins and Minerals” may be included in “3.1. Dietary Factors” section. And there are too many subsections, please combine some of them.

Comments on the Quality of English Language

Please avoid using the first-person “we” narratives for a scientific paper.

Author Response

(The authors gave the same response as above.)

Round 2

Reviewer 1 Report

Comments and Suggestions for Authors

The Authors modified the manuscript according to suggestions.

Comments on the Quality of English Language

Minor editing needed